# MicroRNA-26b Attenuates Platelet Adhesion and Aggregation in Mice

**DOI:** 10.3390/biomedicines10050983

**Published:** 2022-04-23

**Authors:** Linsey J. F. Peters, Constance C. F. M. J. Baaten, Sanne L. Maas, Chang Lu, Magdolna Nagy, Natalie J. Jooss, Kiril Bidzhekov, Donato Santovito, Daniel Moreno-Andrés, Joachim Jankowski, Erik A. L. Biessen, Yvonne Döring, Johan W. M. Heemskerk, Christian Weber, Marijke J. E. Kuijpers, Emiel P. C. van der Vorst

**Affiliations:** 1Institute for Molecular Cardiovascular Research (IMCAR), RWTH Aachen University, 52056 Aachen, Germany; lipeters@ukaachen.de (L.J.F.P.); cbaaten@ukaachen.de (C.C.F.M.J.B.); smaas@ukaachen.de (S.L.M.); jjankowski@ukaachen.de (J.J.); erik.biessen@mumc.nl (E.A.L.B.); 2Interdisciplinary Center for Clinical Research (IZKF), RWTH Aachen University, 52056 Aachen, Germany; 3Department of Pathology, Cardiovascular Research Institute Maastricht (CARIM), Maastricht University, 6200 Maastricht, The Netherlands; c.lu@maastrichtuniversity.nl; 4Institute for Cardiovascular Prevention (IPEK), Ludwig-Maximilians-Universität München, 80337 Munich, Germany; kiril.bidzhekov@med.uni-muenchen.de (K.B.); donato.santovito@gmail.com (D.S.); yvonne.doering@med.unibe.ch (Y.D.); christian.weber@med.uni-muenchen.de (C.W.); 5Department of Biochemistry, Cardiovascular Research Institute Maastricht (CARIM), Maastricht University, 6200 Maastricht, The Netherlands; m.nagy@maastrichtuniversity.nl (M.N.); n.jooss@maastrichtuniversity.nl (N.J.J.); jwmheem722@outlook.com (J.W.M.H.); marijke.kuijpers@maastrichtuniversity.nl (M.J.E.K.); 6Institute of Cardiovascular Sciences, College of Medical and Dental Sciences, University of Birmingham, Edgbaston, Birmingham B15 2TT, UK; 7DZHK (German Center for Cardiovascular Research), Partner Site Munich Heart Alliance, 80337 Munich, Germany; 8Institute for Genetic and Biomedical Research (IRGB), Unit of Milan, National Research Council, 20090 Milan, Italy; 9Department of Biochemistry and Molecular Cell Biology, Medical School, RWTH Aachen University, 52056 Aachen, Germany; dmoreno@ukaachen.de; 10Swiss Cardiovascular Center, Division of Angiology, Inselspital, Bern University Hospital, University of Bern, CH-3010 Bern, Switzerland; 11Synapse Research Institute, Kon. Emmaplein 7, 6217 Maastricht, The Netherlands; 12Munich Cluster for Systems Neurology (SyNergy), 81377 Munich, Germany; 13Thrombosis Expertise Center, Heart and Vascular Center, Maastricht University Medical Center, 6229 Maastricht, The Netherlands

**Keywords:** thrombosis, platelets, microRNAs, microRNA-26b, cardiovascular diseases

## Abstract

Platelets are key regulators of haemostasis, making platelet dysfunction a major driver of thrombosis. Numerous processes that determine platelet function are influenced by microRNAs (miRs). MiR-26b is one of the highest-expressed miRs in healthy platelets, and its expression in platelets is changed in a diseased state. However, the exact effect of this miR on platelet function has not been studied yet. In this study, we made use of a whole-body knockout of *miR-26b* in ApoE-deficient mice in order to determine its impact on platelet function, thrombus formation and platelet signalling both ex vivo and in vivo. We show that a whole-body deficiency of *miR-26b* exacerbated platelet adhesion and aggregation ex vivo. Additionally, in vivo, platelets adhered faster, and larger thrombi were formed in mice lacking miR-26b. Moreover, isolated platelets from miR-26b-deficient mice showed a hyperactivated Src and EGFR signalling. Taken together, we show here for the first time that miR-26b attenuates platelet adhesion and aggregation, possibly through Src and EGFR signalling.

## 1. Introduction

Haemostasis can be considered as the normal response of the blood and blood vessel to injury and is a complex process in order to prevent haemorrhage [1,2]. During this process, platelets, which are anucleated cells derived from megakaryocytes [3], play a critical role. In short, in physiological conditions, platelets flow through the vessel in a quiescent state until a vascular insult or injury occurs. The initial stage of the response to injury involves the recruitment of platelets to the subendothelial matrix via interaction of the platelet surface receptor complex glycoprotein (GP)Ib/IX/V with von Willebrand factor (VWF), which is a glycoprotein present in the plasma that binds to collagen once this is exposed to the blood upon injury to the vessel. Additionally, platelets can bind to collagen in the subendothelial matrix via surface receptors GPVI and integrin α2β1 resulting in a more firm and stable platelet attachment to the site of injury [4,5]. During this initial phase, the platelets are activated due to exposure to local prothrombotic factors such as thromboxane A_2_ and thrombin. Activated platelets recruit additional platelets from the circulation in order to form a platelet plug. To prevent further blood loss, the platelet plug is then stabilized through the generation of fibrin fibres and clot retraction [6].

While haemostasis is the normal response of the vessel to injury, a shift in the haemostatic balance can lead to a hyper- or hypocoagulable state resulting in thrombosis or bleeding, respectively [7]. Both pathological clotting and bleeding disorders are often driven by platelet dysfunction, which can be either inherited or acquired later in life. Systemic diseases, such as chronic kidney disease (CKD) and systemic lupus erythematosus, can lead to acquired platelet dysfunction due to for example uremic toxins or platelet destruction by auto-antibodies [8,9]. Overall, in both haemostasis and pathological thrombosis or bleeding, platelets are one of the key players involved. Platelet function is critical for maintaining the balance of the pro- and anticoagulant state of the blood in which numerous processes are taking part. Many of these processes are regulated by microRNAs (miRs).

MiRs are short, non-coding RNA molecules that regulate post-transcriptional gene expression. By binding to their target messenger RNA (mRNA), they can either repress translation or completely degenerate the mRNA, which, in both scenarios, leads to a decreased expression of the target gene. MiRs usually have various targets and can, thereby, play a critical role in many cellular processes [10,11,12]. Since platelets are anucleated, they need to make use of post-transcriptional regulatory mechanisms in order to regulate all important physiological processes. In line with this, platelets contain a large abundance of miRs, which they inherit directly from the megakaryocytes, further suggesting that miRs play a key role in regulating platelet function also in humans [13,14,15]. For this reason, miRs have been the subject of many studies surrounding platelet development and function. Although the role of various miRs in platelets has already been elucidated, the role of many others remains elusive and, therefore, an interesting focus point for further investigations. One of these interesting, but still rather unexplored, miRs is miR-26b.

MiR-26b has mostly been studied in the field of oncology. For example, it could be demonstrated that miR-26b-5p is downregulated in hepatocellular carcinoma [16], while elevated levels of this miR have been shown in bladder cancer [17]. Besides being linked to carcinogenesis, miR-26b-5p has also been demonstrated to be increased in human atherosclerotic plaque tissue compared to healthy *Arteria mammaria interna* [18], suggesting it may also play an important role in cardiovascular diseases (CVDs). Notably, several studies have highlighted the link between miR-26b and blood platelets. In a study by Nagalla et al., it was unveiled that miR-26b-5p was the second-highest-expressed miR in human leukocyte-depleted platelets from healthy subjects [14]. Furthermore, in patients with sepsis, CKD or diabetes type 2 (DM2), platelet-specific miR-26b-5p expression was lower compared to that in healthy controls, while platelets from patients with polycythaemia vera showed increased levels of miR-26b-5p [19,20,21,22].

The abovementioned findings suggest that miR-26b may play a critical role in regulating processes involved in platelet function and, thereby, influence homeostatic as well as pathological processes. However, so far, no studies have been conducted to investigate the exact role of miR-26b in platelets and platelet function. In order to fill this knowledge gap, we made use of a whole-body miR-26b knockout mouse model, which has previously been described in [23], and aimed to investigate the effects of miR-26b on platelet function using ex vivo and in vivo approaches.

## 2. Materials and Methods

### 2.1. Animals

The whole-body *Apoe^-/-^Mir26b^-/-^* mice were generated as described before [23]. *Apoe^-/-^* mice were used as control, and all mice were on a C57BL/6J background for more than 10 generations. Male and female twelve-week-old mice were used for ex vivo whole-blood perfusion, and four- to five-week-old mice on a chow diet were used for in vivo thrombus formation. All animal experiments were approved by the local authorities (Landesamt für Natur, Umwelt und Verbraucherschutz Nordrhein-Westfalen, Germany, approval number 81-02.04.2019.A363) and complied with the German animal protection law.

### 2.2. Whole-Blood Perfusion Experiments

Whole-blood perfusion experiments were performed without coagulation as described [24]. In short, blood from twelve-week-old *Apoe^-/-^* and *Apoe^-/-^Mir26b^-/-^* mice on a chow diet was isolated retro-orbitally and collected in heparin (5 U/mL) (Sigma-Aldrich, Saint Louis, MO, USA), D-phenylalanyl-L-prolyl-L-arginine chloromethyl ketone (PPACK) (40 µM) (Santa Cruz Biotechnology, Dallas, TX, USA) and fragmin (50 U/mL) (Pfizer, New York, NY, USA). Subsequently, the blood was perfused for 3.5 min at 1000 s^−1^ through a parallel-plate flow chamber, which contained a coverslip coated with collagen type I derived from equine tendon (100 µg/mL; Takeda, Hoofddorp, The Netherlands) (Figure 1A). Platelet activation characteristics were determined by post-perfusion with Tyrode HEPES buffer pH 7.45 (5 mM HEPES, 136 mM NaCl, 2.7 mM KCl, 0.42 mM NaH_2_PO_4_, 1 mg/mL glucose, 1 mg/mL BSA, 2 mM CaCl_2_, 2 mM MgCl_2_ and 1 U/mL heparin) containing AF647-conjugated annexin A5 (1:200) (Molecular Probes, Life Technologies, New York, NY, USA), FITC-conjugated α-CD62P mAb (1:40) (Emfret Analytics, Eibelstadt, Germany) and PE-labelled JON/A mAb (1:20) (Emfret Analytics, Eibelstadt, Germany) to stain for phosphatidylserine (PS) exposure, P-selectin expression and integrin αIIbβ3 activation, respectively. Brightfield and fluorescent images were captured using an EVOS inverted fluorescence microscope (Life Technologies, Carlsbad, CA, USA). Microscopy images were analysed using specific scripts in the open-access Fiji software (Laboratory for Optical and Computational Instrumentation, University of Wisconsin-Madison, WI, USA) as described [24,25]. Multiple thrombus formation parameters were obtained from brightfield images as follows: morphological score of platelet adhesion and thrombus formation (scale: 0–5); surface area coverage of adhered platelet (% SAC); platelet aggregate contraction score (scale: 0–3); platelet aggregate multilayer score (scale: 0–3) and coverage of multi-layered platelet aggregation (% SAC). From the fluorescence images, platelet activation parameters were obtained by determining integrin αIIbβ3 activation (% SAC); P-selectin expression (% SAC) and phosphatidylserine (PS) exposure (% SAC).

### 2.3. Thrombus Formation In Vivo

In vivo thrombus formation experiments were performed in four- to five-week-old mice as described [26]. In short, *Apoe^-/-^* and *Apoe^-/-^Mir26b^-/-^* mice were anaesthetized by intraperitoneal injection of ketamine (0.1 mg/g body weight) and xylazine (0.02 mg/g body weight) and administered with 0.1 µg DyLight488-labelled α-GPIbβ (Emfret Analytics, Eibelstadt, Germany) per gram body weight via tail-vein injection. The mesenteric arterioles and proximate venules were exposed and prepared to be fat-free on a polymer plate with high optical quality. Vessel wall damage was induced by topical application of a filter paper soaked with FeCl_3_ (200 mg/mL) for 3 s, after which fluorescence images were recorded every 5 s via intravital imaging. For the intravital imaging, we used a Ti2-Eclipse microscope (Nikon) equipped with an X-light spinning disk, a camera Zyla 4.2 (Andor) and a LED light engine SpectraX (Lumecor). A Plan Apo λ 10× (NA 0.45) or Plan Apo λ 20× (NA 0.75) air objective was used for acquisition, while the mice were maintained at 37 °C by an OKOLAB UNO-T-H-CO2 top stage incubator. The objective, illumination and acquisition parameters were optimized for each recorded field and were mediated by the Nikon Elements Software in widefield-fluorescence mode. Recording was stopped after 30 min or after full-vessel occlusion (Figure 2A). For each vessel, the time until observation that the first platelet adhered, the first thrombus formed, the thrombus size reached the full diameter of the vessel and the time until the blood vessel was fully occluded were measured. Additionally, we measured the fluorescence signal relative to total vessel area when the first platelets adhered, when the final image was recorded prior to occlusion and at each minute starting from first thrombus formation to, maximally, 15 min, which were all quantified using Fiji software (Laboratory for Optical and Computational Instrumentation, University of Wisconsin-Madison, Madison, WI, USA).

### 2.4. PamGene Kinase Array

Tyrosine kinase profiles were determined using the PamChip^®^ peptide tyrosine kinase microarray system on PamStation^®^12 (PTK; PamGene International, ’s-Hertogenbosch, The Netherlands). Each PTK-PamChip^®^ array contains 196 individual phospho-site(s) that are peptide sequences derived from substrates for tyrosine kinases. Each peptide on the chip builds a 15-amino-acid sequence representing a putative endogenous phosphorylation site, which functions as a tyrosine kinase substrate. The phosphorylation of the peptides is visualized by detection of the fluorescence signal, which is emitted as a result of the binding of the FITC-conjugated PY20 anti-phosphotyrosine antibody.

Blood from *Apoe^-/-^* and *Apoe^-/-^Mir26b^-/-^* mice was collected on acidic citrate dextrose (ACD, 80 mM trisodium citrate, 52 mM citric acid and 180 mM glucose; 1:6 *v*/*v*) by retro orbital puncture. Platelet-rich plasma (PRP) was obtained from the ACD anticoagulated whole blood by centrifugation at 290× *g* for 3 min. A quick spin at 655× *g* was performed to remove the remainder of the erythrocytes from the PRP. For the preparation of washed platelets, PRP was centrifuged at 1960× *g* for five minutes after adding ACD (1:15 *v*/*v*) and apyrase (0.1 U/mL) (Sigma-Aldrich, Saint Louis, MO, USA) to prevent residual platelet activation during the PRP centrifugation step. Subsequently, the pelleted platelets were resuspended in Tyrode HEPES buffer (5 mM HEPES, 136 mM NaCl, 2.7 mM KCl, 0.42 mM NaH_2_PO_4_, 2 mM MgCl_2_) supplemented with 0.1% glucose. For the kinase activity array experiments, platelet count was adjusted to 400 × 10^6^/mL. Platelets supplemented with 2 mM CaCl_2_ were activated 1:10 with thrombin (4 nM f.c.) (Enzyme Research Laboratories Inc., South Bend, IN, USA). After 15 min of activation, platelets were lysed 1:1 using M-PER lysis buffer containing 1:50 protease inhibitor cocktail and 1:50 PhosSTOP (Thermo Fisher Scientific, Waltham, MA, USA). Lysates were centrifuged for 15 min at 10,000× *g* at 4 °C in a pre-cooled centrifuge. Protein quantification was performed using the NanoDrop One (Thermo Fisher Scientific, Waltham, MA, USA). Per array, 4.0 µg of protein was applied for the PTK assay (N = 3 per condition). The assay was carried out according to the standard protocol. The protocol and all reagents were supplied by PamGene International B.V. For the PTK Basic Mix, freshly frozen lysate was added to 4 µL of 10 × protein PTK reaction buffer (PK), 0.4 µL of 1 M dithiothreitol (DTT) solution, 0.4 µL of 100 × bovine serum albumin (BSA), 4 µL of 4 mM ATP, 4 µL of 10 × PTK additive and 0.6 µL of monoclonal anti-phosphotyrosine FITC-conjugate detection antibody (clone PY20). Distilled water was used to adjust the total volume to 40 µL. Before loading the PTK Basic Mix on the array, a blocking step was performed. For this, 30 µL of 2% BSA was applied to the middle of every array. After washing with PTK solution, 40 µL of PTK Basic Mix was applied to each PamChips^®^ array. Subsequently, the microarray assays were run for 94 cycles (passages of the buffer through the array). Images were recorded by a CCD camera, PamStation^®^12, at kinetic read cycles 32–93 at 10, 50 and 200 ms and an end-level read cycle at 10, 20, 50, 100 and 200 ms. The spot intensity at each time point was quantified (and corrected for local background) using BioNavigator software version 6.3 (PamGene International, ’s-Hertogenbosch, the Netherlands).

Serine-threonine kinase profiles were determined using the PamChip^®^ Ser/Thr Kinase assay (STK; PamGene International, ’s-Hertogenbosch, the Netherlands). Each STK-PamChip^®^ array contains 144 individual phospho-site(s) that are peptide sequences derived from substrates for Ser/Thr kinases. Protein lysates from the platelets were prepared as described above. For the STK assay, 2.0 µg of protein and 400 µM ATP were applied per array (N = 3 per condition) together with an antibody mix to detect the phosphorylated Ser/Thr. Subsequently, the sample was pumped back and forth through the porous material for an hour (30 °C) to maximize binding kinetics. Then, the phosphorylation signal was detected after adding a FITC-conjugated antibody. Images were recorded using a LED imaging system, and for each time-point, the spot intensity was quantified using BioNavigator software version 6.3 (PamGene International, ’s-Hertogenbosch, the Netherlands).

A functional scoring method, called upstream kinase analysis (UKA) [27], was used to rank kinases based on specificity scores (which are based on peptides linked to a kinase, derived from 6 databases) and sensitivity scores (based on the differences between knockout vs wildtype condition). Over-representation analyses (ORAs) of the Kyoto Encyclopedia of Genes and Genomes (KEGG) database for the kinases with significant differences from baseline were performed using the ClusterProfiler R-package [28]. The list of kinases of interest contains the kinases with higher median final scores (>1.2). The *p*-values were adjusted for multiple comparisons by the false discovery rate (FDR).

Pathway enrichment analysis was performed on the differentially expressed kinases (median final scores >1.2 and adjusted *p*-value <0.05) with R packages ReactomePA [29]. The kinases present in the platelet-activation KEGG pathway, as well as in our dataset, were selected. The biological complexities in which these kinases may belong to multiple annotation categories were visualized with a network plot using the R package Enrichplot [30].

### 2.5. Statistical Analysis

Data are expressed as mean ± standard error of the mean (s.e.m). Statistical analysis was performed using GraphPad Prism version 9.1.1 (GraphPad Software, Inc., San Diego, CA, USA). Outliers were identified using the ROUT = 1 method. Gaussian distribution was tested via the D’Agostino–Pearson omnibus normality test, while homogeneity of variance was tested using Levene’s test. Significance was tested using either Student’s *t*-test (with Welch correction as required) or Mann–Whitney *U*-test for normally and non-normally distributed data, respectively. A 2-tailed *p*-value <0.05 was considered statistically significant.

## 3. Results

### 3.1. Whole Blood from miR-26b-Deficient Mice Shows Increased Platelet Deposition and Clustering Ex Vivo

Previously, we have described the general blood cell counts in the *Apoe^-/-^Mir26b^-/-^* mice compared to the control [23]. Interestingly, there was no significant difference in white blood cell (4.77 ± 0.35 vs. 4.42 ± 0.36 × 10^3^/mm^3^), red blood cell (9.06 ± 0.23 vs. 8.46 ± 0.15 × 10^6^/mm^3^) and platelet count (1.19 ± 0.14 vs. 1.08 ± 0.11 × 10^6^/mm^3^) between genotypes. In order to establish the effect of miR-26b on platelet function, we made use of an ex vivo whole-blood flow chamber model. Blood was collected from chow-fed *Apoe^-/-^* and *Apoe^-/-^Mir26b^-/-^* mice and perfused through a parallel-plate flow chamber containing a collagen I-coated coverslip (Figure 1A), which allowed us to determine the effect of miR-26b on collagen I-mediated platelet adhesion [31]. Interestingly, blood from *Apoe^-/-^Mir26b^-/-^* mice showed a significantly higher platelet deposition and platelet aggregation, compared to the blood of control mice (Figure 1B,C). Further quantification showed that thrombus height, thrombus contraction, thrombus morphology, integrin activation and P-selectin expression did not differ between both groups (Figure 1D–I). Surprisingly, a significant decrease of PS exposure on miR-26b-deficient platelets was observed compared to control (Figure 1J). Taken together, these results suggest that miR-26b influences platelets by interfering with platelet adhesion and aggregation.

### 3.2. Mice Deficient in miR-26b Have Faster Initial Platelet Adhesion and Increased Final Thrombus Coverage In Vivo

To further investigate the effect of miR-26b on platelets and thrombus formation, we deployed an in vivo model making use of FeCl_3_-induced thrombosis in the mesenteric microcirculation [26] (Figure 2A). Since thrombi formed in arteries are rich in aggregated platelets, as opposed to the relatively platelet-poor thrombi in venous thrombi [32], our analysis focused on arterial thrombus formation. We quantified several parameters including the time until the first platelets adhered to the damaged vessel, which interestingly showed that platelet adhesion occurred a striking 2 min earlier in *Apoe^-/-^Mir26b^-/-^* mice compared to *Apoe^-/-^* control mice (Figure 2B). The times until the first thrombus was formed or until the thrombus size reached the full diameter of the blood vessel were both not affected by miR-26b (Figure 2C,D). Although occlusion time also did not differ between *Apoe^-/-^* and *Apoe^-/-^Mir26b^-/-^* mice, the final size of the thrombus relative to the total vessel area right before occlusion was significantly increased upon miR-26b deficiency (Figure 2E,F). Overall and in line with our ex vivo model, these results clearly demonstrate that miR-26b delays platelet adhesion in vivo and decreases final thrombus size.

### 3.3. Murine miR-26b-Deficient Platelets Have Increased Kinase Activity

In order to elucidate the potential underlying mechanisms fuelling the effects of miR-26b on platelets, we performed a kinase activity array to screen for the effects of miR-26b deficiency on the phosphorylation of tyrosine and serine-threonine kinases in activated murine platelets. Interestingly, the array showed that a total of 47 kinases were significantly more activated in the isolated platelets of *Apoe^-/-^Mir26b^-/-^* mice compared to the platelets of *Apoe^-/-^* controls (Figure 3A). Notably, all kinases important for key platelet receptor signalling, i.e., Src, Syk, Fyn, Fgr and Lyn [31], were significantly more active in knockout platelets compared to control upon thrombin stimulation. Interestingly, thrombin is known to activate platelets via protease-activated receptor-4 (PAR-4), which ultimately signals downstream via Fyn, Lyn and integrins, consequently activating Src. It should be noted though that PAR-4 first signals through several other factors before activating the Src kinase family members [33]. When the differentially activated kinases were plotted into a CORAL kinome tree, we observed that the tyrosine kinase family and, in particular, the Src kinase family were remarkably highlighted (Appendix A), further suggesting that these kinases and their signalling cascades are strongly influenced by miR-26b.

To obtain more insight into the biological processes and pathways that are affected by miR-26b deficiency, we performed an over-representation analysis (ORA) of KEGG for the kinases that were significantly hyperactivated. The analysis showed that PI3K-Akt and MAPK signalling pathways, which are both important in platelet function [34,35], are the top two enriched pathways (Figure 3B). Notably, platelet activation was also amongst the biological processes in which miR-26b plays an important role. Figure 3C visualizes the enriched genes involved in the platelet activation pathway, further underlining the value of Src signalling in our dataset. Taken together, the enrichment analysis further highlighted that miR-26b influences important signalling pathways that are relevant for platelet function.

The kinase activity array makes use of phosphorylation sites of peptides coated on chips in order to predict which kinases differ in activity, enabling the evaluation of the exact peptides that are differentially phosphorylated in the presence of miR-26b. The protein tyrosine kinase (PTK) array (Figure 3D) indicated that six peptides were significantly more phosphorylated in miR-26b-deficient platelets compared to the control samples. Intriguingly, the epidermal growth factor receptor (EGFR), the activation of which has been shown to lead to platelet activation [36], was the most significantly changed (LFC: 1.88) peptide in miR-26b-deficient platelets. Furthermore, in the serine/threonine kinase (STK) array (Figure 3E) two peptides, notably APC/C-CDH1 modulator 1 (ACM1) and retinoblastoma-like protein 2 (RBL2), were significantly more phosphorylated in platelets lacking miR-26b.

Overall, these kinase array data highlight the signalling pathways that are specifically influenced by miR-26b in murine platelets and are, therefore, most likely responsible for the observed effects on platelet function and thrombosis that we observed in vivo.

## 4. Discussion

In this study, we made use of mice that lacked miR-26b (*Apoe^-/-^Mir26b^-/-^*) in order to study the impact of this miR on platelet function and thrombus formation. We could demonstrate that a whole-body deficiency of miR-26b results in increased single-platelet deposition and clustering on a collagen I-coated surface ex vivo, but a decreased PS exposure on a collagen I-dependent surface. In line with this, we observed a faster initial adhesion time and relatively bigger thrombus coverage prior to occlusion in an in vivo model for arterial thrombus formation. In line with this, a kinase array analysis of isolated platelets showed that especially the Src-related kinases and EGFR, which are important for platelet activation and function, have an increased activity in the absence of miR-26b. Collectively, these results suggest that miR-26b inhibits platelet adhesion and aggregation in mice, potentially by influencing Src and EGFR signalling.

Previously, it could be shown that, miR-26b is one the highest-expressed miRs in healthy platelets in humans [14]. However, in a diseased setting, the expression levels of miR-26b have been shown to change in either direction [19,20,21]. For example, one study that focused on the platelet transcriptome of CKD patients demonstrated that the expression of miR-26b-5p was lower in uremic patients, while its levels were again restored in patients who received dialysis [21]. They subsequently showed that the expression levels of three predicted targets of miR-26b, i.e., 15-hydroxyprostaglandin dehydrogenase (HPGD), COP9 signalosome subunit 2 (COPS2) and ubiquitin-specific peptidase 15 (USP15), were also restored upon dialysis. USP15 is known to be involved in the signalling pathway of transforming growth factor-β (TGF-β), which is a key factor in kidney fibrosis, and promotes the activation of TGF-β [37]. In this case, miR-26b affects genes that are critical for CKD development, whereby miR-26b is negatively associated with disease progression. Another study, by Fejes et al., focused on platelets from DM2 patients [19], demonstrating that DM2 patients also show a reduced level of miR-26b-5p and its pre-miR in platelets compared to healthy subjects. Furthermore, inhibition of miR-26b in megakaryocytes in vitro in a hyperglycaemic environment resulted in upregulation of P-selectin mRNA, which again suggests that low miR-26b expression can alter platelet function and stimulate disease progression [19]. Interestingly, Szilágyi et al. also showed that miR-26b levels were reduced in the platelets of septic patients, which was accompanied by elevated P-selectin levels [22]. Notably, our results show that an miR-26b deficiency does not lead to an increased P-selectin expression in the whole-blood perfusion experiment (Figure 1I). However, Fejes et al. [19] demonstrated the effects on mRNA levels of P-selectin and not the platelet surface expression of P-selectin. Although Szilágyi et al. [22] did show the platelet surface expression of P-selectin, this study was performed in septic patients, which is a different disease setting than our flow experiments and is, therefore, not fully comparable. Nonetheless, it remains interesting to note that the increased platelet aggregation (Figure 1C) is not accompanied by an increase in integrin and P-selectin expression (Figure 1H,I). This would indicate that other factors might be influenced by the miR-26b deficiency that increase platelet aggregation. Even more thought provoking is the observation that we see no difference in integrin and P-selectin expression (Figure 1H,I), but do see a decrease in PS exposure (Figure 1J). With regard to the decreased PS-exposure in the whole-blood perfusion experiments, we speculate that this could be due to a shift in the different platelet populations that are present in a thrombus [38,39]. Since PS exposure is also a different process than integrin and P-selectin expression, it could be that the miR-26b interferes in the PS exposure signalling pathway directly and/or affects anoctamin-6 activity [40], but does not interfere with integrin and P-selectin signalling. This should, however, be confirmed in future studies. In contrast to these three studies, which show that, in a diseased subject, platelet-specific expression of miR-26b-5p is lower, a study focusing on polycythaemia vera, a clonal haematopoietic stem cell disorder due to a mutation in Janus kinase 2 (*JAK2*), showed that miR-26b-5p is upregulated in diseased platelets [20]. However, this study did not further elaborate on the possible targets or underlying mechanisms of miR-26b-5p and its potential effect on platelet function in this specific condition. Overall, these studies show that platelet-specific miR-26b-5p levels are generally lower in diseased settings, leading to adversely altered platelet function and gene expression. This correlates very well with our findings in which an absence of miR-26b leads to an adverse effect on platelet function and thrombus formation.

Although the studies that focused on platelet miR-26b-5p in DM2 and CKD patients elucidated some targets of the miR, the studies did not focus further on signalling pathways influenced by miR-26b-5p. Therefore, our findings that the EGFR-peptide is more phosphorylated by knockout platelets is a novel and intriguing observation. Previously, it has been shown that EGF-stimulated EGFR autophosphorylation in human platelets contributes to platelet activation and aggregation [36]. Moreover, ex vivo adhesion of human platelets to collagen was increased when stimulated with EGF [36], which is perfectly in line with our data in which platelet adhesion was increased in miR-26b-deficient platelets that show a hyperphosphorylation of EGFR. Further supporting our findings is the observation that mice injected with EGF show an increased initial platelet deposition in FeCl_3_-induced thrombosis in carotid arteries [36]. Additionally, EGFR signalling activates the PI3K-Akt and MAPK signalling pathways downstream [41], which were interestingly also the top two pathways enriched in miR-26b-deficient platelets in our study. However, since only one study up until now has shown the link between EGFR and platelet activation, a note of caution should be made here, and further studies should validate these findings. Besides the increased phosphorylation of EGFR, we could show that the Src family kinases were also more activated in platelets isolated from miR-26b-deficient mice. Src is mostly responsible for initiating downstream αIIbβ3 signalling, which in turn is essential for platelet adhesion and aggregation [33]. Furthermore, Syk has been shown to be a key player in shear-induced platelet aggregation and thrombus formation under flow [42], which, therefore, correlates very well with our in vivo findings. Further supporting our in vivo findings is the observation that Fyn and Lyn activity is increased, since both have also been previously shown to induce platelet aggregation and granule secretion [43]. Another interesting kinase that has an increased activity in miR-26b-deficient platelets is AMP-activated alpha 1 (AMPKα1). AMPKα1 has been shown to modulate thromboxane and granule secretion in response to collagen [44]. Finally, IGR1R also induced platelet aggregation and granule secretion in response to GPVI-dependent platelet activation [45]. On a side note, an increased kinase activity could either indicate that more kinases are present or that the kinases have a higher level of activity. Overall, these results indicate that the phenotypical change we observe in our mouse model can very likely be explained due to a stronger activation of the EGFR and possibly due to a stronger activated Src signalling cascade.

In our study, we have demonstrated the importance of miR-26b in platelet function. However, it is important to note that the mechanisms involved in platelet function are not influenced only by miRs but rather by a complex interplay of several factors. During the initial phase of the platelet plug formation, i.e., adhesion, platelets are attracted to the area of injury to cover the exposed collagen and other proteins of the subendothelial matrix. Simultaneously, VWF is released by endothelial cells, which allows proper adhesion to collagen and other platelets. Following adhesion, platelets are activated by collagen via platelet surface receptors, leading to morphological changes of the platelet and the release of granules by platelets. These granules contain important factors for further activation and platelet aggregation, including fibrinogen, VWF, platelet activation factor, ATP and ADP. In the plasma, prothrombin is then converted by coagulation factors to its active form thrombin, which in turn promotes the activation and aggregation of platelets via PARs and also catalyses the conversion of fibrinogen to fibrin [4,5,6]. The aforementioned cascade shows that a great variety of factors play a role in the mechanisms involved in haemostasis and thrombosis. MiRs can interact with several of these factors, and a single factor can be targeted by several miRs, thereby influencing mechanisms that lead to thrombosis. Another interesting aspect to highlight is that, although platelets are anucleated, functional miR processing exists in platelets as members of the cytoplasmic miR processing complex have been reported in platelets [13]. However, an abundance of mature miRs over pre-miRs has been previously reported [13], suggesting that platelets inherit most of their mature miRs from the megakaryocyte or take up the mature miRs from the plasma directly. Since our mouse model is a whole-body knockout, the deficiency could exert its effects on the platelets via several pathways, i.e., during miR processing inside the platelet, uptake of the miR from the plasma or via inheritance from the megakaryocyte. Most likely though, the deficiency already plays a part in the megakaryocyte, which would produce aberrant platelets in this case.

It is important to note that our findings are only based on a murine model and have not been validated in a human study yet. Human and mouse platelets have inherent differences, such as the lack of certain surface receptors in murine platelets including the thrombin receptor PAR-1 [46]. Hence, our findings need to be validated further in a human setting before drawing final conclusions on the general effect of miR-26b in platelets. Additionally, since the mouse model is whole-body deficient, we cannot exclude the influence of miR-26b on other cell types besides platelets, e.g., endothelial cells, leukocytes and erythrocytes, that could affect in the in vivo thrombus formation (Figure 2), which should be taken as a note of caution. Nevertheless, our data do demonstrate the effect of miR-26b on platelet function. Namely, during the ex vivo whole-blood flow experiment (Figure 1), the influence of the endothelium is excluded, and the kinase activity array (Figure 3) is performed on isolated platelets, thus excluding all other cell types.

To conclude, our study demonstrates that a whole-body deficiency of miR-26b in mice results in increased platelet adhesion and aggregation, possibly due to increased Src and EGFR signalling. These findings might be interesting to study further in other pathologies in which platelets play an important role, such as CVDs. The main underlying cause for CVDs is atherosclerosis, wherein platelets are known to promote its initiation and acceleration [47]. As mentioned before, miR-26b levels were increased in atherosclerotic plaque tissue [18], further suggesting that future research could focus on elucidating its role in CVDs and atherosclerosis. Moreover, future studies could validate our findings in human platelets and, as miR-26b is highly conserved between species [48], we expect to see similar results as in our mouse model. Lastly, future studies should also focus on determining potential targets of miR-26b through which it exerts its effects on platelet function. Overall, our data show that reduced miR-26b levels seem to be related to an increased platelet response. Therefore, restoring miR-26b levels would be an interesting therapeutic approach for thrombosis, and altering miR-26b levels could also potentially be of benefit for related diseases, such as CKD, DM2 and atherosclerosis.

## Figures and Tables

**Figure 1 biomedicines-10-00983-f001:**
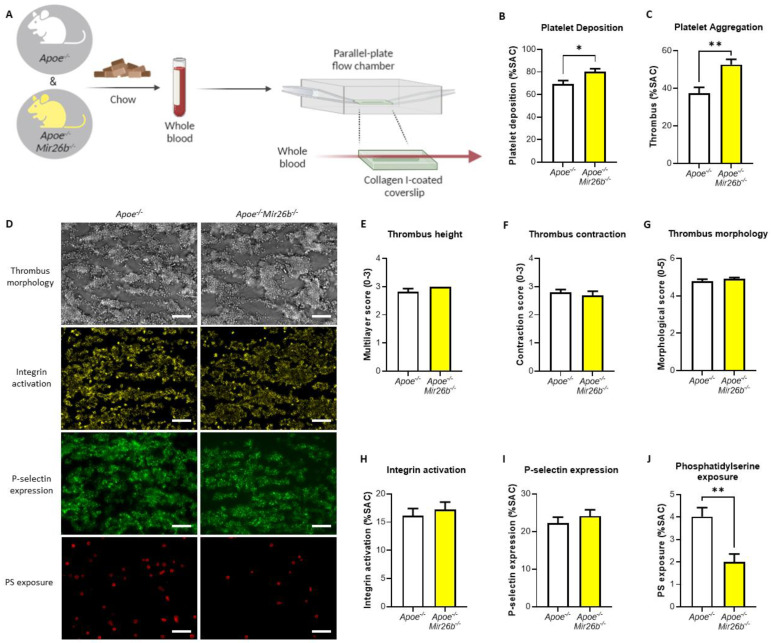
MicroRNA-26b deficiency in mice leads to increased platelet deposition, clustering and decreased phosphatidylserine exposure. (**A**) Schematic representation of the parallel-plate flow chamber experiment. Whole blood from *Apoe^-/-^* and *Apoe^-/-^Mir26b^-/-^* mice on a chow diet was perfused through a flow chamber, which contained a collagen I-coated coverslip. Created with BioRender. (**B**–**J**) (*n* = 9–14) Multiple parameters were assessed via brightfield and fluorescence imaging, including platelet deposition (**B**), thrombus coverage indicating platelet aggregation (**C**), multilayer score (**E**), contraction score (**F**), morphological score (**G**), integrin activation (**H**), P-selectin expression (**I**) and phosphatidylserine (PS) exposure (**J**). (**D**) Representative images of thrombus morphology, PS exposure, P-selectin expression and integrin activation. Scale = 20 µm * *p* < 0.05; ** *p* < 0.01.

**Figure 2 biomedicines-10-00983-f002:**
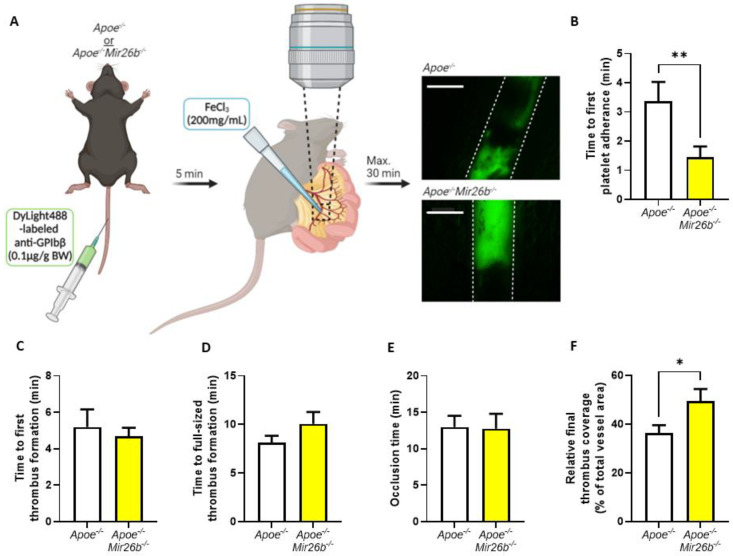
MicroRNA-26b deficiency in mice leads to a faster platelet adhesion and the formation of larger thrombi. (**A**) Schematic overview of the experimental setup. *Apoe^-/-^* and *Apoe^-/-^Mir26b^-/-^* mice were administered a DyLight488-labelled anti-GPIbβ (0.1 μg/g BW) via tail-vein injection. After 5 min, the mesenteric vasculature was exposed, and the blood vessels were damaged by application of FeCl_3_ (200 mg/mL). The blood vessels were recorded for, maximally, 30 min with intravital microscopy. Representative pictures of relative final thrombus coverage. Dotted lines indicate vessel wall. Scale = 100 µm. Created with BioRender. (**B**) Time point at which the first platelet adhered (*n* = 10–14 vessels from 5 to 8 individual mice). (**C**) Time at which that the first thrombus formed (*n* = 11–15 vessels from 5 to 8 individual mice). (**D**) Time point at which the thrombus size reached the full diameter of the vessel. (**E**) Time point at which the blood vessel was occluded (*n* = 10–11 vessels from 5 to 8 individual mice). (**F**) Fluorescence signal measured relative to total vessel area when the final image was recorded prior to occlusion (*G*, *n* = 10–11 vessels from 5 to 8 individual mice). * *p* < 0.05; ** *p* < 0.01. BW: body weight.

**Figure 3 biomedicines-10-00983-f003:**
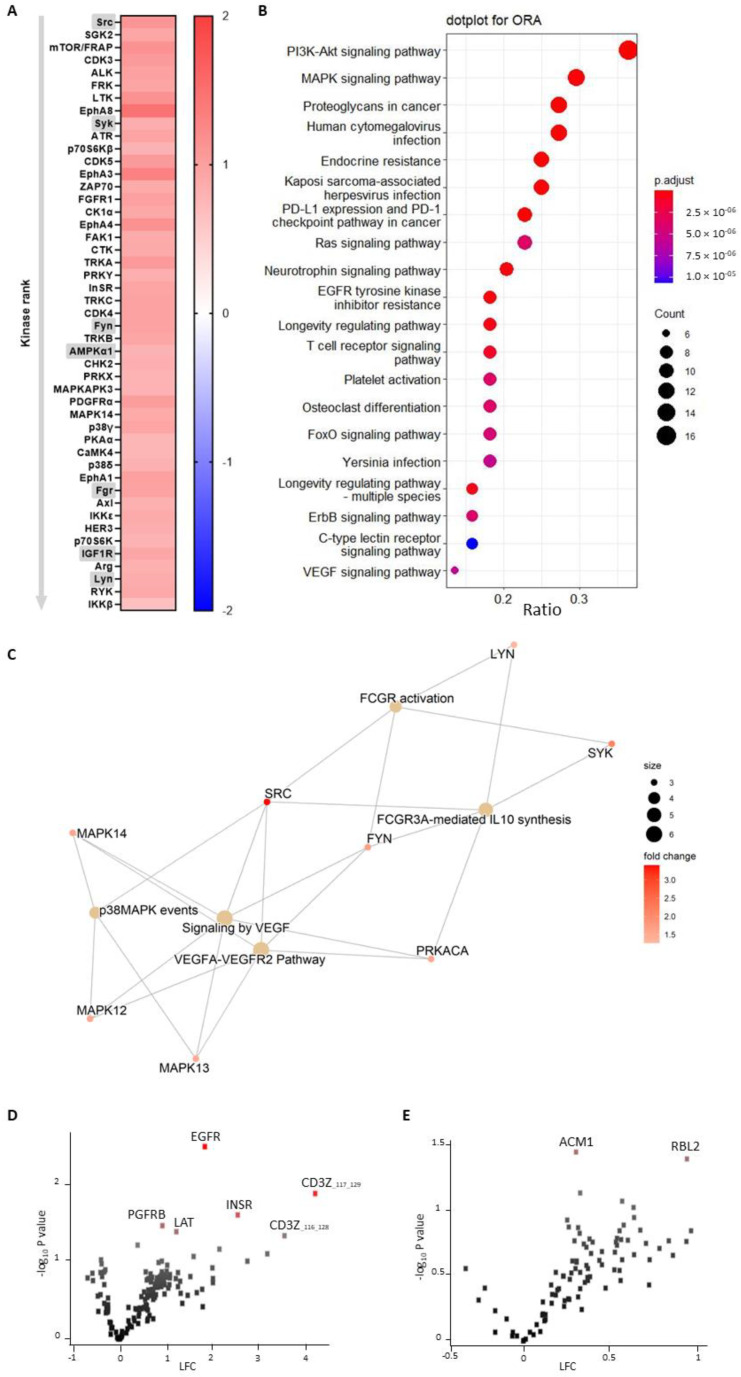
Activated miR-26b-deficient murine platelets show increased kinase activity compared to control platelets. (**A**) Heatmap for kinase activation protein tyrosine kinase (PTK) and serine/threonine kinase (STK) arrays on thrombin-stimulated platelets from *Apoe^-/-^* and *Apoe^-/-^Mir26b^-/-^* mice (*n* = 3). All kinases that were significantly changed in activity are ranked based on their median final score (cut-off value of 1.2). Colour is based on the median kinase statistic, which represents effect size and directionality (red: increased activity; blue; decreased activity). Kinases important for platelet function are highlighted in grey. (**B**) Dot plot for over-representation analysis (ORA) of KEGG with significant differences from baseline using the PTK and STK dataset. (**C**) Pathway enrichment visualizing kinases involved in platelet activation. (**D**,**E**) Volcano plots demonstrating fold change (*Apoe^-/-^Mir26b^-/-^* vs. *Apoe^-/-^*) and *p*-value for peptide phosphorylation from PTK ((**D**), *n* = 3) and STK ((**E**), *n* = 3) arrays. Red dots represent significantly altered phosphopeptides, *p*-value <0.05, paired *t*-test. LFC: log2-fold change.

## Data Availability

Not applicable.

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
