# Peer review of "MicroRNA-26b Attenuates Platelet Adhesion and Aggregation in Mice"

_biomedicines, 2022, doi:10.3390/biomedicines10050983_

Round 1
Reviewer 1 Report
The authors described the association between microRNAs and platelet aggregation and adhesion by investigation in mice. The approach and the data are interesting, and the manuscript is written well. However, there are some points required the revision. The detailed information was shown below.
- The authors suggested nicroRNA was related to platelet aggregation and adhesion and the phenomenon is also related to chronic kidney disease, diabetes type 2 and some diseases. Although the experimental results showed the microRNA affected to thrombus formation and phenomenon, the experiments were performed in mice. In addition, the experiment was performed in one situation, and the situation is different from in vivo. The discussion should be performed with the limitation. Please describe the limitation about the haemostasis and thrombosis mechanism difference between human and mice in the discussion section as the limitation.
- The authors found the importance of microRNA in haemostasis and thrombosis and the results data suggested microRNA was related to the mechanisms. On the other hand, the mechanisms are largely dependent on thrombin. In the discussion section, please describe that there are several important factors in the haemostasis and thrombosis mechanisms and microRNA is one of them.
Reviewer 2 Report
Major:
The authors nicely demonstrate the impact of the absence of miR-26b on platelet function, however they keep away from trying to hypothesize a possible mechanism of action:
- Does an effect of miR in platelets imply that the transcription machinery is active in platelets in a relevant manner?
- Or is the effect of the miR absence on platelets to be found in the MGK’s producing aberrant platelets?
- What does an increased kinase activity mean: more kinase present or kinase more activated? And then what could be the putative target of the miR?
The authors also do not engage in an attempt to explain how in the flow chamber experiments increased platelet aggregation is not accompanied by an increase in activated integrin or P-selectin, and even with a (marked!) decrease in PS exposure.
Minor:
l.53 VWF only being mentioned as "present in the matrix", seems even for an “in short” a bit too short?
L93 “platelet-specific miR-26b-5p was differentially expressed compared to healthy controls”: OK, but why not immediately give whether this was up- or downregulated?
M&M: sources of materials used need to be given: heparin, PPACK, fragmin, collagen type I, annexin A5, anti- CD62P-mAb, apyrase, thrombin….
L152: avoid “advertisement’-like name giving: “Kinase activity profiling” better than “PamGene Kinase Array”?
in 2.4 “Pam” is mentioned 16 times (in the whole manuscript 23 times), with the company address given 4 times…)
L166: what is the purpose of adding an additional 1:15 v/v ACD to PRP that already contained 1:6 v/v (and thus even higher due to hematocrit)? And why not adding apyrase already when collecting the blood?
L168 the Tyrode Hepes buffer composition here differs significantly from the l.119 Tyrode Hepes buffer that in addition contains 1 mg/mL glucose, 1 mg/mL BSA, 2 mM CaCl2, and 1 U/mL heparin…
L186 what is meant with “the microarray assays were run for 94 cycles”: what is cycling here?
L230-231: significant digits?
L236: why would the platelet adhesion to collagen I in the flow chamber be specific for GPVI? No VWF or integrin α2ß1 involved?
As aggregation seems to be more significantly affected then deposition (fig. 1B-C) it seems a bit odd to state l.242 “Taken together these results suggest that miR-26b 242 influences platelets by interfering with platelet adhesion” ?
Fig.1: no explanation is given on how the different parameters have been determined: how is platelet aggregation determined, what is meant by thrombus morphology?
l. 292: “thrombin is known to activate platelets via protease-activated receptor-4 (PAR-4), which signals downstream via Fyn, Lyn and integrins, consequently activating Src [30]. “: Yes, but then really downstream as [30] also states: via Gq, activation of PLC, release of Ca2+ and activation of PKC, and then SFKs… blocking SFKs “attenuates” response to thrombin. Please reformulate more carefully.
l.344 “Fe3Cl treatment, excluding an effect of endothelial cell-specific miR26b” OK, but what about erythrocytes, leukocytes?
(and should be FeCl3)
l.365, l.368 “inhibition of miR-26b in megakaryocytes/…/ resulted in upregulation of P-selectin mRNA, which again suggests that a low miR-26b expression can alter platelet function”: good place to enter and discuss the no changes that were found here in P-selectin expression and integrin activation, and the decrease in PS exposure.
l.386 “miR-26b deficient platelets that show a hyperphosphorylation of EGFR.” : hyperphosphorylation of EGFR actually has not been demonstrated, but of an EGFR-peptide. And all evidence that EGF stimulates platelets is coming from a single paper ref33, which therefore needs sufficient caution.
Round 2
Reviewer 2 Report
The authors did answer my remarks in a satisfactory manner.
Just two small remarks remain:
- collagen type I (Takeda): OK for the commercial source, however please also mention that this is from equine tendon, and how much was used to coat
- L230-231: significant digits? We have included the p-values for white blood cell count, red blood cell count and platelet count (Lines 250-252 of the revised manuscript).
I wasn’t aiming for p-values (indeed no need for them if stated not significant), but I was indicating that the number of digits given e.g. for red blood cell (9061.667 ±229.353 vs. 8461.667 ±150.918 x103 /mm3 ; p = 0.147257) is not significant: something like: 9.06.106 ± 0.23.106 vs. 8.46.106 ± 0.15.106 /mm3 would make more sense.
by the way: the official abbreviation for von Willebrand factor is VWF (not vWF)
